# Insights and Strategies of Melanoma Immunotherapy: Predictive Biomarkers of Response and Resistance and Strategies to Improve Response Rates

**DOI:** 10.3390/ijms24010041

**Published:** 2022-12-20

**Authors:** Attila A. Seyhan, Claudio Carini

**Affiliations:** 1Laboratory of Translational Oncology and Experimental Cancer Therapeutics, Warren Alpert Medical School, Brown University, Providence, RI 02912, USA; 2Department of Pathology and Laboratory Medicine, Warren Alpert Medical School, Brown University, Providence, RI 02912, USA; 3Joint Program in Cancer Biology, Lifespan Health System and Brown University, Providence, RI 02912, USA; 4Legorreta Cancer Center, Brown University, Providence, RI 02912, USA; 5School of Cancer & Pharmaceutical Sciences, Faculty of Life Sciences & Medicine, New Hunt’s House, Guy’s Campus, King’s College London, London SE1 1UL, UK; 6Biomarkers Consortium, Foundation of the National Institute of Health, Bethesda, MD 20892, USA

**Keywords:** immune checkpoint inhibitors, melanoma, responders and non-responders, biomarkers, therapeutic strategies

## Abstract

Despite the recent successes and durable responses with immune checkpoint inhibitors (ICI), many cancer patients, including those with melanoma, do not derive long-term benefits from ICI therapies. The lack of predictive biomarkers to stratify patients to targeted treatments has been the driver of primary treatment failure and represents an unmet medical need in melanoma and other cancers. Understanding genomic correlations with response and resistance to ICI will enhance cancer patients’ benefits. Building on insights into interplay with the complex tumor microenvironment (TME), the ultimate goal should be assessing how the tumor ’instructs’ the local immune system to create its privileged niche with a focus on genomic reprogramming within the TME. It is hypothesized that this genomic reprogramming determines the response to ICI. Furthermore, emerging genomic signatures of ICI response, including those related to neoantigens, antigen presentation, DNA repair, and oncogenic pathways, are gaining momentum. In addition, emerging data suggest a role for checkpoint regulators, T cell functionality, chromatin modifiers, and copy-number alterations in mediating the selective response to ICI. As such, efforts to contextualize genomic correlations with response into a more insightful understanding of tumor immune biology will help the development of novel biomarkers and therapeutic strategies to overcome ICI resistance.

## 1. Introduction

Immune checkpoint inhibitors (ICIs) are revolutionizing the treatment of melanoma and other types of cancers, with 7 agents now approved in the US and a rich pipeline of new agents and mechanisms in development [1,2,3,4]. Notwithstanding the excitement around these developments, there is a significant unmet medical need in the form of patient stratification and therapy resistance. Melanoma affects more than 1 million Americans, and there is an increasing incidence of melanoma worldwide. Approx. 300,000 new cases are diagnosed in the US each year [5], with the average annual cost for treatment estimated at $3.3 billion [6].

Numerous successes have been achieved with anti-PD1, anti-CTLA4, or combination therapies [5,7] in treating melanoma and other types of cancers. The groundbreaking finding by Leach et al. [8] showed that antibodies blocking the T cell co-inhibitory receptor CTLA-4 can augment immune responses against tumor cells in mice. This finding gave rise to ipilimumab, the first ICI to increase the survival of patients with metastatic melanoma [9,10] which was granted FDA approval in 2011 for the treatment of metastatic melanoma.

In 2014, additional T cell immune checkpoint blocking antibodies, anti-PD-1 (pembrolizumab) and anti-PD-L1 (nivolumab) [11], received FDA approval. The combination of anti-PD-1 (pembrolizumab) and anti-PD-L1 (nivolumab) resulted in the augmented survival of patients with metastatic melanoma compared to patients treated with ipilimumab alone or chemotherapy alone [12,13,14].

In 2022, relatlimab, which targets lymphocyte-activation gene 3 (LAG-3), was approved by the FDA for adult and pediatric use with metastatic melanoma [15].

Following those developments, high levels of PD-L1 expression in cancer cells and tumor mutational burden (TMB) have been shown in melanoma to correlate with clinical responses to ICI [13].

Despite these data, a substantial number of melanoma patients fail to respond to ICI, leading to premature death [16,17,18]. Considerable effort has been made to identify biomarkers that predict clinical response/resistance to ICI [19]. Despite the successes of ICI, even with the combination of ipilimumab and nivolumab, the five-year survival for the intention-to-treat (ITT) population was only 53%. This means that 47% of patients do not reach long-lasting benefits and succumb to the disease [20], suggesting the need for predictive biomarkers of response to overcome ICI resistance.

Despite the approval of all of these new ICIs and combination therapy, physicians are nowhere near having predictive biomarkers of response to ICI. Presently, physicians have no idea which patients will or will not respond to ICI in the absence of predictive biomarkers of response.

Recently, Carlino et al. [21] demonstrated that combining anti-PD-1 and anti-CTLA-4 ICI in stage IV melanoma resulted in the highest 5-year overall survival rate of all the other therapies. Increased PD-L1 expression and TMB have been shown to correlate well with melanoma response to ICI [13]. Unfortunately, both PD-1 and CTLA4 are unable to predict the outcome [22]. As a result, we run the risk of undertreating some patients who might benefit from ICI.

To date, there are no reliable predictive biomarkers able to stratify responders and non-responders to ICI. Clinical factors, such as volume and disease sites, serum lactate dehydrogenase levels, and BRAF mutation status, used to select the initial therapy for patients with advanced melanoma have been rather unreliable [21].

The clinical trial data of melanoma response to ICI have identified three almost equal populations of patients: (1) those who respond (responders), (2) those who fail to ever respond (innate resistance), and (3) those who initially respond or have a prolonged period of disease stabilization, but eventually develop disease progression (acquired resistance) [7,16,23,24,25].

The current stratification strategies, including genetic mutations and variations in mutational load, have shown some correlation with response to ICI. However, they are unable in predicting patient response to ICI [26]. It has been shown that tumor cells possess genetic and epigenetic traits that facilitate immune evasion [27]. Tumors can mutate to evade the innate and adaptive immune response [19], rendering ICI therapy ineffective [23,28,29]. ICI resistance can start from different cells and their interactions in the tumor ecosystem [29,30]. Recent studies have revealed insights into the mechanism of ICI resistance [7,29,30]. As a result, ICI therapies elicit significantly limited efficacy in many of the cancer types due to drug resistance and toxicity.

The lack of predictive biomarkers to stratify patients to targeted treatments has been the driver of primary treatment failure and represents an unmet medical need in melanoma and other cancers. Therefore, understanding genomic correlations with response and resistance to ICI will be beneficial to cancer patients.

Here, we review recent studies involving ICI therapies in melanoma and other cancers that have revealed important insights about the tumor immune microenvironment in melanoma and other cancers, the mechanisms supporting these findings, and genomic correlations with response and resistance to ICI.

We further discuss how the insights gained from these studies are guiding more precise analyses of the mechanisms of action of ICI therapy and novel immunotherapy approaches, including novel combination therapies to overcome resistance to ICI therapy and turning “cold tumors” into “hot tumors”.

Consequently, these efforts will help us better understand the mechanisms involved in response and resistance to ICI, thus achieving more effective and durable use of immunotherapy.

## 2. Immune Checkpoint Inhibitors

As illustrated in Figure 1, immunosuppressive mechanisms involving the expression of checkpoint proteins (e.g., PD-1, CTLA-4, LAG-3, TIM-3, and NKG2A/B), activation of cell death programs, and accumulation of various immunosuppressive cells are instrumental in maintaining immune homeostasis and self-tolerance [31,32,33,34]. Cancer cells exploit those immune homeostasis mechanisms to evade the immune system [35]. Several ICI treatments are successful against a wide variety of cancers, given the durability of response and improved side effect profile compared to chemotherapeutic agents. Nonetheless, only a few ICIs have been approved by the FDA [34].

## 3. Melanoma Biomarkers of Response to ICI

Physicians would greatly benefit in therapeutic decision-making by having predictive biomarker(s) of response to stratify melanoma and other types of cancer patients into responders and non-responders. A general representation of predictive biomarkers of response to ICIs in several cancers is shown in Table 1. However, the significant proportion of non-responders and treatment-associated toxicities and drug resistance remain one of the major obstacles to therapeutic success of ICI in melanoma and other types of cancers.

### 3.1. Established Clinical Biomarkers

Different types of biomarkers and their clinical utility in metastatic melanoma, including those approved by the FDA and others still at the experimental stage, are shown in Table 2 along with the clinical setting in which they are used.

As summarized in Table 2, the earliest biomarkers that helped to diagnose the presence of melanoma include human melanoma black-45 (HMB-45), melan-A, tyrosinase, microphthalmia transcription factor (MITF), S100, SM5-1, chondroitin sulfate proteoglycan 4 (CSPG4), loss of p16 protein expression, biomarker panels, and gene arrays. Those biomarkers have been used to screen healthy patients before the diagnosis of melanoma in order to help stratify patients into benign vs. malignant stages of the disease.

The earliest approved clinical biomarkers that helped to inform the prognosis of metastatic melanoma relied on baseline clinical characteristics, such as serum levels of lactate dehydrogenase (LDH), a marker of tumor burden [41]. Elevated serum LDH levels have been demonstrated to be a negative prognostic marker, irrespective of the given treatment [42,43,44,45]. Generally, high LDH levels are associated with poor overall survival (OS) compared with normal LDH levels. Elevated levels of LDH have been used as a biomarker to assess patient staging [46]. For example, it has been shown that among ca. 30% of patients with 4–5-year OS following treatment with BRAF and MEK inhibitors, only a few of those patients had high LDH levels before the initiation of therapy [47].

Likewise, elevated serum levels of S100B have demonstrated to have a prognostic value in both metastatic [48,49] and high-risk resected melanoma settings [50].

Gene expression profiling has shown to have a prognostic value that complements existing biomarkers in patients with melanoma [47,51,52].

In metastatic melanoma, a well-characterized predictive biomarker of response guiding the therapeutic decision process is the BRAF V600 mutation, which is somehow predictive of response to BRAF ± MEK inhibition with low rates of primary resistance [53,54,55]. The response rate to BRAF and MEK inhibitors in metastatic melanoma patients with the BRAF V600 mutation is ca. 70% in selected patients, with less than 10% of patients having the highest response to progressive disease [55,56,57,58].

Several oncogenic driver mutations have been identified as predictive biomarkers of response from targeted agents, including NRAS, NF-1, and c-KIT, that provide insight into the probability of therapeutic response to a specific treatment [59,60,61,62,63]. Presently, the presence of a BRAF V600 mutation is the only validated predictive biomarker for melanoma patients.

### 3.2. Emerging Predictive Biomarkers

As shown in Table 2, many emerging biomarkers are currently being evaluated in clinical studies. Du et al. [64] reported that several genomic and transcriptomic-based biomarkers have been explored as potential predictors of ICI response.

Predictive biomarkers of response include TMB, neoantigen load [7,65,66,67,68,69], HLA-I genotype [70,71], cytolytic activity [72], aneuploidy [73], and T cell repertoire [46], which exhibit high predictability to ICI response. Other predictive biomarkers of response are PDL-1 expression, LAG-3 expression, CD8+ T cells at tumor invasive margin, IL-17 expression, immune-related gene expression signatures, and T-cell receptor (TCR) signature.

Additional biomarkers that are used to assess ICI response include ctDNA profiles, absolute lymphocyte count, proliferating CD8+ T cells, increase of T-cell subsets and checkpoint molecules (PDL-1, LAG-3), granzyme B expression, and T-cell receptor (TCR) signature [37,38,39,40].

Unfortunately, many of those potential biomarkers of ICI response have not yet been validated [74,75,76,77,78].

A recent study by Carter et al. [74] questioned the validity of the immuno-predictive score (IMPRES), a predictor of ICI response in melanoma consisting of 15 pairwise transcriptomic signatures that analyze the relationship between immune checkpoint genes reported by Auslander et al. [76]. The IMPRES is context-dependent and could not reproducibly predict ICI response in the context of metastatic melanoma [76].

Moreover, Xiao et al. [77] questioned the reproducibility of the Immune Cells.Sig [79] signature in melanoma, demonstrating inconsistencies in the prediction capability of ImmuneCells.Sig across different RNA-seq datasets [77]. The performance of the ImmuneCells.Sig signature in predicting ICI outcomes in four melanoma patient datasets, using the same implementation scheme as Xiong et al. [79], showed that there were inconsistencies across different datasets [77].

#### 3.2.1. Gene Expression Signatures

Gene expression signatures (GES) have also been identified as predictive biomarkers of response to ICI and have been validated in several independent datasets (e.g., immune-predictive score, IMPRES consisting of 15 immune genes). IFN-γ-responsive genes were also used to predict ICI response in metastatic melanoma [30,46,72,73,76,80,81,82,83,84,85,86].

A panel of pan-tumor T cell-inflamed GES consisting of 18 IFN-γ-responsive genes was validated and confirmed to predict the response to ICI in pre-treatment tumor specimens from nine types of cancers, including melanoma [83].

MHC-I/II gene signatures have also been explored as predictive biomarkers of ICI response in melanoma [85,87,88].

Carter et al. [74] reported that immuno-predictive score (IMPRES), a predictor of ICI response in melanoma encompassing 15 pairwise transcriptomics relations between immune checkpoint gene [76], did not reproducibly predict the response to ICI in metastatic melanoma.

It was argued that many factors may contribute to the limited successes of those biomarkers, such as: (1) the predictive biomarkers have been derived from pre-clinical studies; (2) evaluation of the biomarkers in clinical specimens only included baseline biopsies and peripheral blood samples; (3) batch effect, lack of reproducibility might have contributed to the failure of the biomarkers for ICI response [74,75,77,78].

To address these issues, numerous researchers have developed predictive biomarkers to reduce batch noise and other technical issues. Expanded predictive biomarker panels have resulted in higher reproducibility as opposed to predictive signatures based on individual biomarkers [89,90,91,92].

Tian et al. [91] reported that the combined BRAF, KRAS, and PI3KCA mutation signature resulted in a favorable predictive response to cetuximab for patients with colorectal cancer [91].

#### 3.2.2. Gene Expression Signatures at Baseline and On-Treatment Tumor Specimens

Genome-wide analysis of transcriptomic and genomic profiles of baseline and on-therapy tumor specimens from patients treated with ICI provides a comprehensive view into the mechanisms underlying tumor response and resistance to ICI [93]. 

Grasso et al. [93] reported that the mechanism of action of ICI is based on the interaction between immune effector cells and cancer cell targets. Tumor studies conducting comprehensive analyses of transcriptomic and genomic profiling have focused not only on the genetic alterations and gene expression profiles of cancer cells [25,30,85,88], but also on the composition of immune infiltrates and expression of immune-activating gene programs [36,46,67,75,80,82,83,85,86,88,94,95,96,97,98].

Du et al. [64] reported that pathway-based signatures derived from on-treatment tumor specimens were predictive of the response to anti-PD1 blockade in patients with metastatic melanoma.

Other studies of breast cancer suggested that post-treatment tumor samples were more informative than pre-treatment samples [99,100,101].

Conversely, Wallin et al. developed adaptive immune signatures based on tumor samples obtained during the early course of treatment, showing that the signatures were highly predictive of the response to ICI in patients with metastatic melanoma [102].

Auslander et al. built an immune-predictive score (IMPRES), which encompasses 15 pairwise transcriptomic relations between immune checkpoint genes, to predict the response of metastatic melanoma to ICB therapy [76].

The IMPRES signature produced better predictive scores with post-treatment samples than with pre-treatment samples in two independent datasets [76].

In support of these findings, a recent proteome profiling study of samples from patients with metastatic melanoma undergoing either tumor infiltrating lymphocyte based or anti-PD1 immunotherapy demonstrated that the fatty acid oxidation pathway was significantly enriched in responder patients. These results underlined the critical role of mitochondrial metabolism, including fatty acid metabolism, in conferring response to immunotherapy [103].

It is now widely accepted that post-treatment tumor specimens are generally much more informative than pre-treatment specimens and may provide more valuable insight into dynamic changes at the transcriptional level that correlate with clinical response, resulting in a higher predictive score.

In conclusion, although pathway signatures derived from post-treatment samples are highly predictive of therapeutic response to anti-PD1 in patients with metastatic melanoma, further studies are warranted to confirm the predictive value of those signatures in larger cohorts of patients with metastatic melanoma.

#### 3.2.3. Pathway Signatures

Du et al. [64] developed pathway-specific signatures in pre-treatment (PASS-PRE) and on-treatment (PASS-ON) tumor specimens based on transcriptomic data and clinical information from a large dataset of metastatic melanoma patients treated with anti-PD1. Both PASS-PRE and PASS-ON signatures were validated in three independent datasets of metastatic melanoma. Compared to existing molecular signatures, it was concluded that the on-treatment (PASS-ON) tumor specimen signature exhibited a robust and better predictive value for metastatic melanoma patients who responded to anti-PD1 across all four datasets. 

The pre-treatment pathway signatures included six pathways for predicting the response to anti-PD1 treatment, including: (1) complement cascade; (2) regulation of insulin-like growth factor IGF transport and uptake by insulin-like growth factor binding proteins IGFBPS; (3) binding and uptake of ligands by scavenger receptors; (4) plasma lipoprotein remodeling; (5) IL2 family signaling; and (6) retinoic acid (RA) biosynthesis pathways. Complement cascade, binding, and uptake of ligands by scavenger receptors and IL2 family signaling pathways are related to immune and inflammation, whereas plasma lipoprotein remodeling and the RA biosynthesis pathways are related to metabolism.

In contrast to the pathway-based signature analysis of on-treatment samples, Du et al. [64] identified four pathways, including: (1) peroxisomal lipid metabolism; (2) generation of second messenger molecules; (3) fatty acid metabolism; and (4) PD1 signaling. Of note, peroxisomal lipid metabolism and fatty acid metabolism are related to fatty acid and lipid metabolism [103]. Generation of second messenger molecules is a central signaling pathway in T-cell receptor (TCR) stimulation. Likewise, PD1 signaling plays an important role in immunoregulation as an immunoregulatory signaling pathway.

To further validate the predictive performance of a pathway-based super signature for on-treatment samples (PASS-ON), Du et al. [64] tested three independent datasets with RNA-seq data available for on-treatment samples, including those of Gide et al. and Lee et al. [96,104], and the MGH cohort demonstrated the effectiveness of PASS-ON in predicting patient response to anti-PD1. Patients with high PASS-ON signature scores were associated with significantly improved PFS compared to those with low signature scores in all tested patients.

Furthermore, Du et al. [64] demonstrated that the time-response interaction pathway-based super signature for pre- and on-treatment samples had reasonable predictive power. The study suggested that pathway-based biomarker signatures derived from on-treatment tumor specimens compared to pretreatment tumor specimens were better predictors of response to anti-PD1 therapies in metastatic melanoma patients.

#### 3.2.4. Tumor Antigens

Huang et al. [22] investigated several melanoma-relevant tumor-specific antigens, cancer germline genes, melanocyte differentiation antigens, overexpressed antigens, neoantigens, neuropeptides, and other sources of immunogenic antigens, such as immunogenic epitopes, have also been explored as novel predictive biomarkers of ICI response to melanoma.

Neoantigens are derived from tumor-specific somatic mutations and are exclusively expressed in cancer cells and absent in normal human tissue. The majority (95%) of somatic mutations are single-nucleotide variants (SNVs), which lead to aberrant protein and peptide expression with single amino acid substitutions [105].

Neopeptides also arise from nucleotide insertions or deletions (indels), leading to the expression of aberrant proteins and peptides with frameshift or non-frameshift sequences depending on the number of nucleotides added and deleted. While the minority of mutations are indels (<5% for melanoma) [106,107], frameshift mutations can generate a number of immunogenic neoepitopes that are highly distinct from the self.

Other sources of immunogenic antigens, including immunogenic epitopes, can also derive from mutations associated with gene fusion, aberrant messenger-RNA splicing with retained introns, or aberrant translation resulting in cryptic antigens, and genomically integrated endogenous retroviral sequences as a result of previous retroviral infections, although they are epigenetically silenced, can be reactivated in tumors [106], as in the case of cancer germline antigens.

Furthermore, tumors often present aberrant patterns of DNA methylation, resulting in the demethylation, ectopic expression, and presentation of cancer germline genes to T cells relevant in immune recognition [106,108].

For example, cancer germline genes such as MAGEA1 and NY-ESO-1 are silenced epigenetically through methylation in human tissue, with the exception of male germ cells and trophoblastic cells, which lack MHC-I molecules.

PRAME (preferentially expressed antigen in melanoma), a member of the cancer-testis antigen family, has been reported to be frequently overexpressed in many cancers, including melanoma, which indicates advanced cancer stages and poor clinical prognosis [109,110]. As such, overexpressed PRAME is a potential immunotherapy target. PRAME-specific immunotherapies are currently in development for many cancers, including melanoma. For example, a recent study demonstrated that uterine carcinosarcoma, synovial sarcoma, and leiomyosarcoma patients would potentially benefit from PRAME-specific immunotherapies [109].

#### 3.2.5. Genomic Alterations

Considerable effort has been made to identify genomic alterations and transcriptome profiles as predictive biomarkers of ICI response. Numerous studies have identified distinct stages of CD8+ T cells linked to positive response or failure to ICI treatment [98]. 

Moreover, tumors from patients responding to ICI showed a higher number of cancer-associated somatic mutations (i.e., mutated antigens or neoantigens) targeted by T cells [65]. The IFN-γ signature has also been shown to predict the response to ICI (e.g., anti-PD-1) in melanoma [111] and in other types of cancers [6,93]. Gene expression signatures obtained from bulk melanoma tumor or single-cell profiling and the TME have been shown to be correlated with sensitivity and resistance to several ICIs [86,88,98,112,113,114].

Collectively, the gene expression signatures associated with response to ICI in metastatic melanoma represent distinct characteristics and play an important function in different signaling pathways, including the inflammatory response, type I interferon signaling pathway, cytokines, and others.

## 4. Who Is Responding to ICI?

Recent data have demonstrated that immunotherapies against immune checkpoints (e.g., CTLA-4 or PD-1) downregulate two main negative regulators of the anti-tumor immune response [93,115,116,117], resulting in durable anti-tumor responses in a subset of cancer patients, including those with melanoma [2,118].

Another key factor contributing to anti-tumor immune response following ICI treatment [93] is the pre-existing level of T cell infiltration of the tumor [119,120,121], representing the immunogenicity of the cancer cells.

Analysis of tumor biopsies from ICI-treated patients showed that clinical responses associated with ICI were mediated by tumor-infiltrating T cells reactivated following ICI treatment [121,122].

A combination of immunohistochemical (IHC) analysis with RNA-seq performed on cancer biopsies from patients treated with anti-CTLA-4 antibody (ipilimumab) before or after treatment with anti-PD-1 antibody (nivolumab) demonstrated that a major response to anti-CTLA-4 requires cancer cells with high levels of MHC-I expression at baseline, whereas the response to anti-PD-1 was more strongly associated with a pre-existing interferon-γ gene expression signature [88].

Liu et al. demonstrated that the response to anti-PD-1 therapy (with or without prior anti-CTLA-4 treatment) was associated with increased MHC-I and MHC-II expression [85]. This study demonstrated that patients not responding to therapy have occasional genetic alterations in antigen presentation genes [85].

Biopsies from patients with metastatic melanoma treated with anti-PD-1 monotherapy (nivolumab) in part 1 of the CheckMate 038 study showed an increase in immune cell subtypes with elevated immune activation gene signatures seen in responders to therapy [25].

The transcriptome analysis of tumor biopsies from patients treated with anti-PD-1 monotherapy (nivolumab) or in combination anti-PD1 plus anti-CTLA-4 therapy (ipilimumab) correlated well with the in vitro analysis of gene expression signatures of melanoma cell lines following exposure to interferon-γ [93].

It appears that cancer cells become enablers of the immune response via the expression of IFN-γ-response genes, triggering the upregulation of antigen presentation, amplification of the interferon response, and induction of chemokines (i.e., CXCL9 and CXCL10) to entice immune cells to the TME. Thus, T cell-induced IFN-γ correlates with ICI therapy response [93]. Collectively, the degree of the anti-tumor T cell response and downstream IFN-γ signaling are the main drivers of response or resistance to ICI therapy [93].

However, little is known about how tumor-intrinsic loss of IFN-γ signaling impacts TILs. The question remains whether tumor-intrinsic IFN-γ signaling actively regulates the infiltration or function of TILs?

Shen et al. [123] demonstrated that IFN-γR1 knockout melanomas and IFN-γR1KO melanomas in B6 mice had reduced infiltration and function of tumor-infiltrating lymphocytes (TILs). Furthermore, long-distance effects of IFN-γ on tumor cells also play a crucial role in anti-tumor immunity [123]. These recent findings revealed an important role of tumor-intrinsic IFN-γ signaling and IFN-γ-response genes in shaping TILs.

## 5. Tumor-Immunity Cycle and Resistance Mechanisms Involving Tumor Immunophenotypes

As discussed in the literature [124] and illustrated in Figure 2 and Figure 3, the anti-tumor-immunity cycle is a gradual process mediated to a large extent by CD8+ T lymphocytes and involves a multi-step process.

In the immune-desert phenotype, immune cells are absent from the tumor and its periphery. In the immune-excluded phenotype, immune cells accumulate but do not efficiently infiltrate. In the immune-inflamed phenotype, immune cells infiltrate but their effects are inhibited. Notably, the three different phenotypes have different response rates to immune checkpoint inhibitors.

Based on the spatial distribution of cytotoxic immune cells in the TME, a tumor is classified as an immune-inflamed, immune-excluded, or immune-desert phenotype (Figure 2 and Figure 3) [126]. Immune-inflamed tumors (i.e., “hot tumors”) are characterized by increased T cell infiltration, high interferon-γ (IFN-γ) signaling, elevated expression of PD-L1, and increased TMB [127]. Inflamed tumors are more responsive to ICIs than non-inflamed tumors [119,128]. Immune-deserted tumors (i.e., “cold tumors”), on the other hand, exhibit characteristics where CD8+ T lymphocytes localize only at the tumor margin and do not infiltrate the tumor [129]. In immune-desert tumors, CD8+ T lymphocytes are absent from the actual tumor and its periphery [129]. In addition to poor T cell infiltration, “cold tumors” display low mutational load, decreased MHC class I expression, and reduced PD-L1 expression [127].

Cold tumors also harbor immunosuppressive cells, including T-regulatory cells (Tregs) and myeloid-derived suppressor cells (MDSCs), and tumor-associated macrophages (TAMs) are key sources of many of these inhibitory factors [127]. As a result, cold tumors lack innate immunity or the innate antitumor immune features present in cold tumors’ may be ineffective due to the lack of immune cells [126]. The three tumor immune phenotypes have different response rates to ICIs and cold tumors respond poorly to ICI monotherapy [119].

### 5.1. Tumor Cell-Intrinsic and Tumor Cell-Extrinsic Resistance Mechanisms

Many factors play a role in T cells driving tumor resistance, ultimately leading to a noninflamed T cell phenotype and failed antitumor immunity (Figure 3 and Figure 4).

Several other resistance mechanisms involving both tumor cell-intrinsic and tumor cell-extrinsic sources have been described (Figure 3 and Figure 4 and Table 3) [34]. In the case of tumor cell-intrinsic mechanisms, a lack of neoantigen development, impaired antigen presentation, and other primary factors contribute to the resistance to immunotherapy [130,131,132,133,134,135,136]. Tumor cell-extrinsic mechanisms encompass increased recruitment and activity of inhibitory immune cells within the TME and upregulation of LAG-3 and TIM-3 [16,47,137,138,139,140].

### 5.2. Immune Resistance Mechanisms in Melanoma

As illustrated in Figure 2, Figure 3, Figure 4 and Figure 5 and Table 3, resistance to ICI therapy is one of the most significant challenges to achieving a durable tumor response in many types of cancers [16,24,34,182,183].

Several mechanisms, including primary (lack of tumor response to initial immunotherapy) and secondary/acquired resistance to cancer immunotherapy (initial response to tumor followed by a lack of response) have been described as the major resistance mechanisms to ICI therapy [16] (Figure 2, Figure 3, Figure 4 and Figure 5 and Table 3).

As for melanoma, Huang et al. [22] demonstrated that several key mechanisms are involved in the immune resistance to ICI, such as immune tolerance, T cell exhaustion, immune cell-mediated immunosuppression, expression of immune checkpoint ligands, and tumor escape.

Immune tolerance describes the lack of response by the immune system to substances or antigens that have the potential to induce an immune response. Immune tolerance to an individual’s own antigens occurs through both central and peripheral tolerance mechanisms. While central tolerance occurs via thymic deletion of high-affinity auto-reactive T cells, peripheral tolerance is maintained by other mechanisms (e.g., suppression by Treg cells and anergy) and induced by many mechanisms, including sub-optimal T cell co-stimulation, deletion via apoptosis, or conversion into Treg cells. The dose of antigen and TCR affinity are considered to be the major drivers of these mechanisms [22]. Notably, both the lymph node and tumor environments blunt T-cell effector functions and offer a rationale for the failure of tumor-specific responses to effectively counter tumor progression [186].

T cell exhaustion is a specific T cell differentiation process mediated by chronic antigen stimulation, which leads to increased expression of co-inhibitory immune receptors that are presumed to decrease chronic TCR signaling and regulate activation-induced cell death.

In this state of simultaneous TCR stimulation and co-inhibitory pathway stimulation, exhausted T cells (T_EX_ cells) exhibit reduced effector functions (i.e., cytokine production and proliferative potential), but can survive in the hostile TME. Notably, T cell exhaustion appears to be a dynamic and progressive process that includes intermediate reversible states more permissive to stimulation by ICI.

Cell-mediated immunosuppression involves immunosuppressive cells, including MDSCs, Treg cells, and tolerogenic DCs, instructing effector T and B cells to not respond to positive immune stimuli.

Upregulation of immune checkpoint ligands PD-L1 and PD-L2 is often seen in melanoma and other cancers in response to robust inflammatory signals as a homeostatic mechanism adopted by cancer cells to shield themselves from immune attack. Interaction of PD-L1 and PD-L2 ligands through binding to PD-1 receptors expressed on tumor-specific T cells initiate a negative signaling process downstream of PD-1, which reduces T cell activation and impairs tumor-killing function [11]. As a consequence, expression of PD-L1 or PD-L2 in melanoma cells can neutralize the positive T cell signals mediated by the MHC-I and MHC-II antigen presentation process.

Tumor escape is a mechanism that cancer cells utilize to escape from anti-tumor immunity and immune surveillance. Tumor escape can be mediated by tumor-extrinsic mechanisms in the TME and by the tumor itself, which can evolve to evade immune recognition. Under strong immune selective pressure, heterogeneous tumor cells can result in clonal evolution, selection, and enrichment of specific tumor cells that can evade immune recognition, leading to immunotherapy resistance. The evasion of immune recognition by these treatment-resistant cells can take place via inactivation of antigen-presentation processes (e.g., B2M, HLA, TAP, etc.) and/or IFN-γ-response genes (e.g., JAK1, JAK2).

## 6. Therapeutic Strategies to Turn “Cold Tumors” into “Hot Tumors”

Several strategies have been investigated to elucidate the mechanisms underlying how T cells are driven into “hot tumors” in order to improve the efficacy of ICI therapy (Figure 6 and Table 4) [124]. Several clinical trials have tested these novel therapeutic modalities as interventions in combination with ICI to overcome ICI monotherapy resistance and attempt to turn “cold tumors’ into “hot tumors” (Table 5).

Grasso et al. [93] showed that a robust anti-tumor immune response relies on the interplay of key factors that can be modulated with innovative interventions.

Recent preclinical and clinical findings have provided insight into the immunological implications of canonical cancer signaling pathways (e.g., WNT-beta-catenin signaling, cell cycle regulatory signaling, mitogen-activated protein kinase signaling, and pathways activated by loss of the tumor suppressor phosphoinositide phosphatase PTEN), thus providing new opportunities for the development of new treatments for those patients who do not respond to ICI monotherapies [218].

Combined therapeutic strategies from preclinical [115,117,219] and clinical studies [14] have shown that anti-PD-1 plus anti-CTLA-4 (nivolumab and ipilimumab) treatments elicit stronger immune stimulation in stage IV melanoma than monotherapy alone, resulting in a favorable anti-tumor immune response. Response to ICI, either following anti-CTLA-4 monotherapy or in combination with anti-PD-1 therapy, triggers a robust T cell response that generated an appreciable antitumor response [218].

Another approach to turn “cold tumors” into “hot tumors” is through the intra-tumoral delivery of oncolytic viruses or Toll-like receptor agonists capable of inducing intra-tumoral interferon production, which triggers the pattern recognition pathways with consequent boosting of the anti-PD1 immune response rate [187,190,220].

Recently, different approaches have been used to boost the response to ICI. Activation of the STING pathway [221], inhibition of immune suppressive factors (e.g., WNT signaling or the adenosine pathway) [111,209,222,223], as well as the release of other immune checkpoints (e.g., LAG-3, TIM-3, or TIGIT, etc.) in T cells [111] have been explored, but no favorable clinical outcomes have yet been reported in patients.

Altogether, these new therapeutic strategies provide new opportunities for cancer immunotherapy for patients who do not respond to ICI [111].

## 7. Lessons Learned: ICI Therapies in Melanoma

Huang et al. [22] reported that several factors, including the immune TME, tumor-associated immune cells, and different host factors, contribute to the ICI resistance. Melanoma resistance has armed us with a handful of information that can be applied to other types of cancers, such as: (1) the ability to present cancer antigens through MHC-I and elevated TMB; (2) tumor-antigen-specific T cells play a crucial role in the response to ICI; (3) reactivation of terminally exhausted T cells could be considered a biomarker for PD-1 blockade, which is detectable as early as one week after ICI dosing; (4) melanoma immunosuppressive mechanisms are complicated and need additional research to remodel their interaction, cooperation, and dynamics during tumor progression and in immunotherapy resistance; and (5) Treg cells are emerging as a key mechanism of resistance to PD-1 blockade, but not necessarily CTLA-4 blockade; (6) emerging neoadjuvant immunotherapy trials are anticipated to provide new insight into pharmacodynamic immune responses and advance the development of rational immunotherapy and neoadjuvant combination regimens while avoiding toxicity and significantly improving patient management; (7) longitudinal assessment of pre- and on-treatment patient specimens is required to determine prognostic vs. predictive use of immune and other parameters, including genomic parameters correlating with patient outcomes, and deduce their biologic role in response to ICI therapy based on their modulation during treatment; and last but not least (8) melanoma-specific oncogenic programs supporting metabolic plasticity and fitness, together with clinical and preclinical evidence of differential activity of ICI therapy depending on the tumor metabolic state, should provide new research opportunities to evaluate these relationships as potential biomarkers for patient stratification and treatment allocation, and formulate novel precision-medicine combinations depending on metabolic and immune therapies.

## 8. Conclusions and Future Perspectives

Even with a high level of immunogenicity, metastatic melanoma grows and spreads rapidly via escape mutations and immunosuppression. Although combination ICI therapies targeting CTLA-4 and PD-1 can efficiently target some ICI-resistant mechanisms by improving T cell priming, re-activating PD-1 high CD8 + exhausted T cells, and reversing Treg suppression, many patients still do not derive durable clinical benefit.

Therefore, with the availability of novel immunotherapeutic agents, the mechanism of single-agent therapies needs to be better characterized in order to guide effective rational ICI combinations. In addition, since the immunologic effects of ICI therapy occur early, we need to focus on these early events to identify specific biomarkers, mechanisms of resistance, and neoadjuvant therapies.

Furthermore, toxicity from current and new ICI combinations remains a critical step to be addressed [224]. Because of concerns regarding adverse events from current and new ICI combinations, insight gained from molecular mediators of immune toxicity (e.g., antibody- vs. T cell-mediated) resulting from combination ICI therapies would help in avoiding these side effects and the development of novel combination ICI therapies to other cancers.

As a result, new efficacy and toxicity data deriving from different ICI therapies across various tumor types will help to characterize key parameters for predicting ICI response, thus limiting toxicity and enhancing therapeutic decision processes to overcome ICI resistance. In fact, new efficacy and toxicity data from many ICI therapies across various tumor types are helping in the characterization of key parameters for predicting response, limiting toxicity, and informing therapeutic decisions to overcome resistance.

Furthermore, the identification of novel genomic correlations with resistance to ICI requires well-annotated data from diverse patient cohorts and tumor histology to detect rare response-associated variants [159,225]. With a greater focus on detailed genomics as well as epigenomics, proteomics, and metabolomics, microbiome and biomarker studies will better characterize the relationships among host immunity, tumor biology, and mechanisms of resistance and response to ICI. Ultimately, molecular and clinical data from these studies will have to be adequately integrated and blended into preclinical studies.

For this reason, systems biology, computational biology, biostatistics, as well as machine learning and artificial intelligence approaches must be coordinated to integrate and help interpret this large dataset, thus translating the findings into actionable novel biomarkers that capture the complexity of multiple alterations affecting response and therapeutic success.

In addition, several other key challenges face ICI therapy that must be addressed to move the field ahead [226], including: (1) development of preclinical models that translate to human immunity; (2) identification and validation of the dominant drivers involved in cancer immunity; (3) deeper insights into organ-specific immune TME; (4) exploration of differences in molecular and cellular drivers between primary and secondary immune escape; (5) characterization of the benefit of endogenous vs. synthetic immunity; (6) investigation of ICI therapy combinations with other drugs (Figure 6, Table 4 and Table 5) in early-phase clinical studies; (7) investigation of steroids and immune suppression on immunotherapy and autoimmune adverse events and toxicities; (8) boosting personalized medicine approaches by using composite biomarkers; (9) developing and implementing refined regulatory endpoints for immunotherapy; and (10) optimizing durable survival with a combination of multi-agent immunotherapy regimens [226].

Consequently, all of these efforts will result in a better understanding of the mechanisms involved in response and resistance to ICI, hence facilitating the development of biomarkers and novel therapies.

In conclusion, these efforts should result in a deeper quantitative and conceptual understanding of the mechanisms involved in ICI response and resistance, thus facilitating the development of rational biomarkers and therapies.

## Figures and Tables

**Figure 1 ijms-24-00041-f001:**
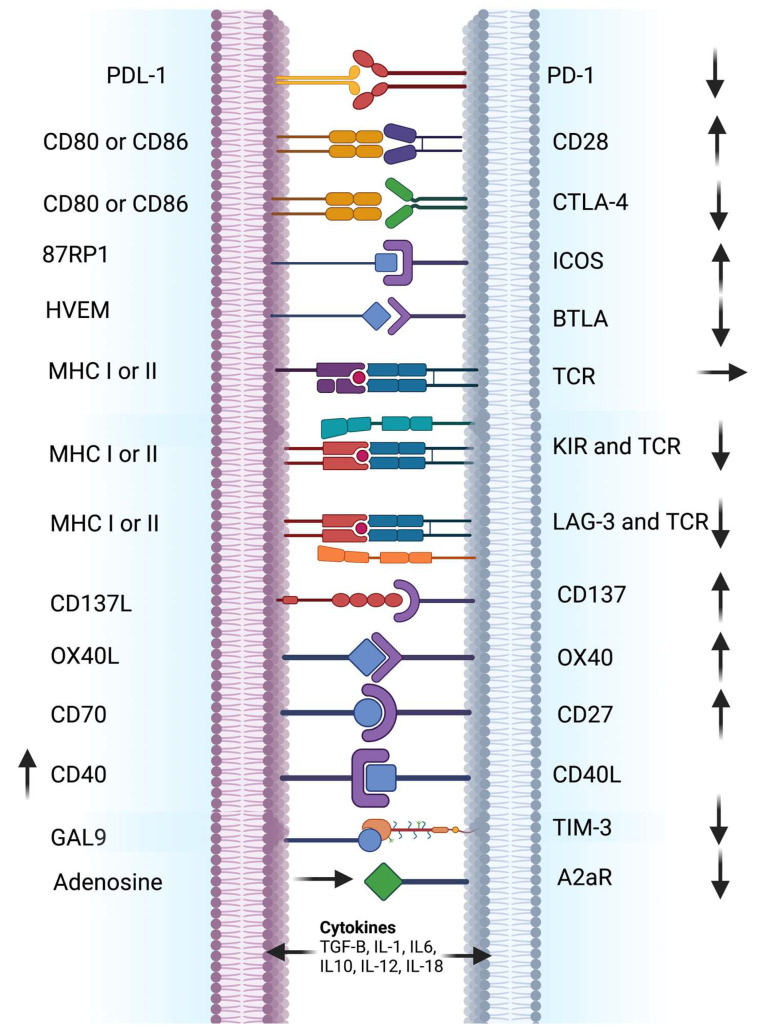
Immune cell and cancer cell or antigen-presenting cell receptor-ligand interactions involved in immune checkpoint modulation. The figure illustrates an overview of implicated receptor-ligand interactions and their general effects on the immune response. Several immune cells, such as CD8 + T cells, CD4 + T cells, NK cells, and Tregs, express specific receptors, which are recognized and bound by their specific ligands present on the surface of various cancer cells or antigen-presenting cells. The illustration also depicts examples of different receptors and ligands involved in ICI modulation, along with generalized stimulatory (↑) or inhibitory (↓) effects. ICIs block various inhibitory receptor-ligand interactions leading to the activation of immune cells, which leads to tumor regression. ICI therapies, including anti-PD-1/PD-L1 and anti-CTLA-4, have shown clinical efficacy for many cancers, which provided opportunities for developing alternative ICIs (e.g., anti-LAG-3, anti-TIM-3, and anti-NKG2). Adapted from ref. [3,34]. Created with BioRender.com (accessed on 18 November 2022). Definitions of abbreviations are found in the abbreviations list at the end of the text.

**Figure 2 ijms-24-00041-f002:**
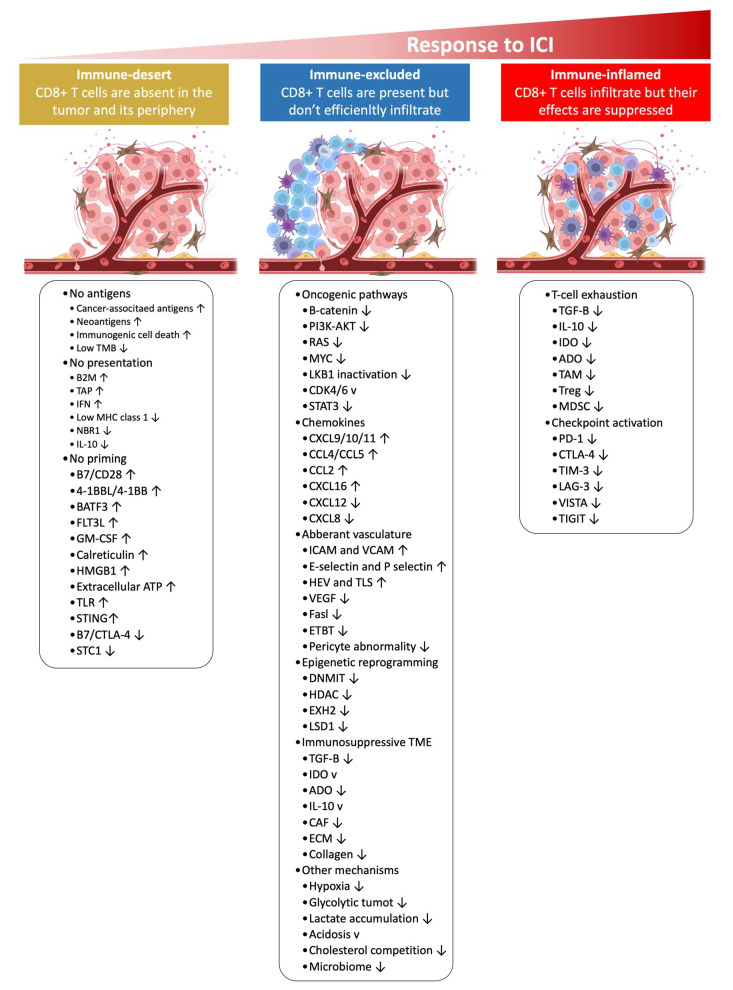
Mechanisms of distinct tumor phenotypes, including immune-inflamed, immune-desert, and immune-excluded tumors, are associated with specific inhibitory and stimulatory biological mechanisms. Immune-inflamed tumors are permissive to immune cell infiltration; however, immune cells in the TME can be suppressed due to checkpoint activation. On the other hand, immune-desert tumors may be devoid of T cell priming due to the lack of tumor antigens, defective antigen processing and presentation processes, or impaired interactions between dendritic cells and T cells. The immune-excluded tumors, on the other hand, may display aberrant chemokines, activated oncogenic pathways, hypoxia, aberrant vasculature, or an immunosuppressive TME (e.g., stromal barriers). Adapted from ref. [124]. ↓ (downward arrow), inhibitors; ↑ (upward arrow), stimulatory factors. Created with BioRender.com (URL accessed on 18 November 2022). Definitions of abbreviations are found in the abbreviations list at the end of the text.

**Figure 3 ijms-24-00041-f003:**
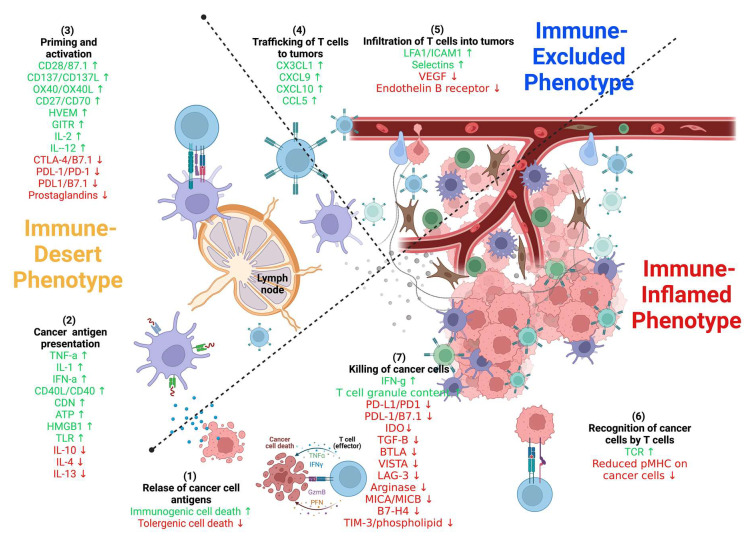
**Stimulatory and inhibitory factors in the cancer-immunity cycle.** Each step of the cancer-immunity cycle requires the coordination of several stimulatory and inhibitory factors. Stimulatory factors shown in green with the upward arrow (↑) indicating promotion of immunity. Inhibitors shown in red with the downward arrow (↓) help keep the process in check and reduce immune activity and/or prevent autoimmunity. Examples of such factors and the primary steps at which they can act are shown. Antitumor immunity is mediated predominantly by CD8+ T cells and tumor immunity involves: (1) tumor antigen release, (2) tumor antigen processing and presentation by APCs (e.g., dendritic cells), (3) priming and activation of T cells, (4) trafficking of T cells via the bloodstream to tumors, (5) infiltration of activated T cells into the tumor parenchyma from the vasculature or tumor periphery, (6) recognition of tumor cells through antigenic peptide-MHC complexes on the surface of tumor cells by T cells, and (7) killing of tumor cells by cytotoxic T cells through granule exocytosis and release of perforin and granzyme or through the Fas/FasL pathway by inducing ferroptosis and pyroptosis. The release of additional antigens from dead tumor cells allows the continuation of the tumor-immunity cycle. Tumors with the immune-desert phenotype cannot pass steps 1–3 due to the absence of T cells in both the tumor and its margins. On the other hand, tumors with the immune-excluded phenotype cannot pass steps 4–5 due to the absence of T cells in the tumor bed. On the other hand, tumors with the immune-inflamed phenotype cannot pass steps 6–7 because of T cell exhaustion and checkpoint upregulation. Immune checkpoint proteins, such as PD1 and CTLA-4, can suppress the development of an active immune response by acting primarily at the level of T cell development and proliferation (step 3). Adapted from ref. [124,125]. Created with BioRender.com (URL (accessed on 18 November 2022). Definitions of abbreviations are found in the abbreviations list at the end of the text.

**Figure 4 ijms-24-00041-f004:**
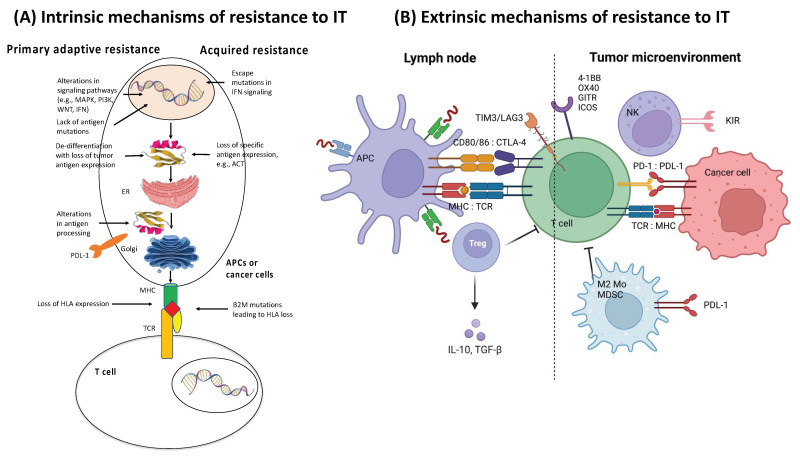
Mechanisms of resistance to immunotherapy. (**A**) Intrinsic mechanisms of resistance to immunotherapy. Examples of intrinsic mechanisms of adaptive resistance involve altered signaling pathways, limited mutational burden, de-differentiation of tumor resulting in loss of neoantigen expression, defective antigen processing, constitutive PD-L1 expression, and loss of HLA expression. Examples of intrinsic mechanisms of acquired resistance include loss of antigenic target, loss of HLA expression, and escape mutations in IFN signaling. (**B**) Extrinsic mechanisms of resistance to ICI therapy (right panel). Examples of extrinsic mechanisms of resistance involve upregulated or constitutive immune checkpoint expression, immunosuppressive cytokine release (e.g., CSF-1, TGFβ, adenosine) within the TME, T cell exhaustion and phenotypic switching, as well as elevation of immunosuppressive cell populations (e.g., Treg, MDSC, and M2 macrophages). Adapted from ref. [34]. Created with BioRender.com (URL (accessed on 18 November 2022). Definitions of abbreviations are found in the abbreviations list at the end of the text.

**Figure 5 ijms-24-00041-f005:**
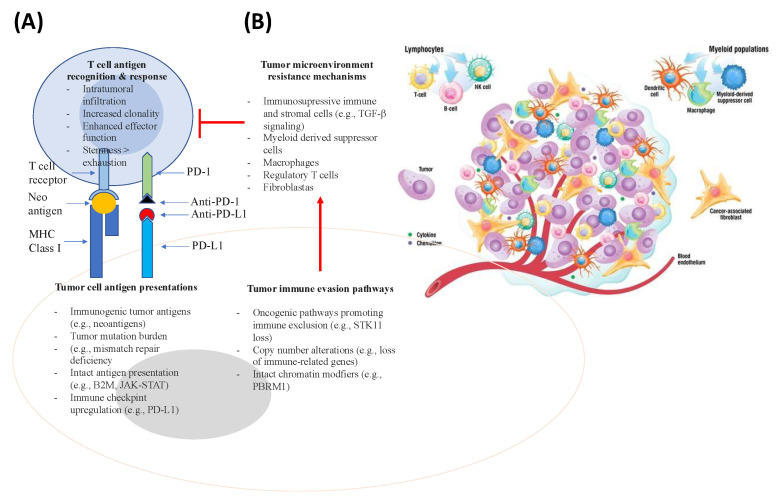
Genomic correlations with response and resistance to ICI therapy within the immune TME. The panel (**A**) represents correlations with response, focusing on antigen presentation and recognition. The panel (**B**) represents resistance pathways that promote tumor immune evasion mechanisms that induce immunosuppressive cells, leading to the inhibition of T cell-mediated anti-tumor response (right side). Adapted from [113]. The panel (**B**) also illustrates the TME, which consists of cellular and non-cellular components. The cellular component consists of cancer cells, endothelial cells, pericytes, carcinoma-associated fibroblasts, and immune cells. The immune compartment comprises many immune cell populations (e.g., T cells, B cells, natural killer cells, tumor-associated macrophages, myeloid-derived suppressor cells, and dendritic cells). The non-cellular component of the TME, on the other hand, is represented by the extracellular matrix and functions as a scaffold. Components of the TME interact via the extracellular matrix, cell-cell contacts, and through the release of cytokines, chemokines, extracellular vesicles, and others. Adapted from ref. [113,184,185]. Definitions of abbreviations are found in the abbreviations list at the end of the text.

**Figure 6 ijms-24-00041-f006:**
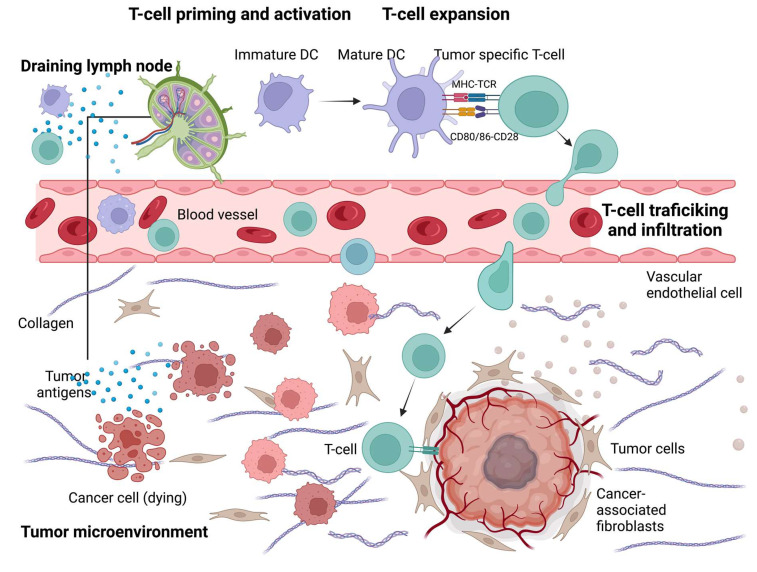
**Approaches to turn a “cold tumor” into a “hot tumor”.** Some representative therapeutic strategies to increase T cell infiltration and improve the efficacy of ICI. T cell priming and activation: Oncolytic viruses, local thermal ablation therapy (e.g., radiofrequency ablation), chemotherapy, and radiotherapy are all capable of inducing immunogenic cell death to promote T cell priming and activation. Local administration of immune adjuvants, such as TLR agonists, promotes the activation of DCs. Epigenetic modification inhibitors can promote T cell priming by increasing the expression of tumor antigens and by restoring antigen processing and presentation mechanisms. T cell expansion: Cancer vaccines and adoptive cellular therapies, such as CAR-T cells, can promote the expansion of tumor-specific T cells. T cell trafficking and infiltration: Intrinsic oncogenic pathway inhibitors, antiangiogenic therapies, TGFβ inhibitors, CXCR4 inhibitors, and epigenetic modification inhibitors can promote T cell trafficking and enable T cells to infiltrate the tumor more effectively. Adapted from ref. [124]. Created with Briorender.com (URL (accessed on 18 November 2022). Definitions of abbreviations are found in the abbreviations list at the end of the text.

**Table 1 ijms-24-00041-t001:** A general representation of predictive biomarkers of response to immune checkpoint inhibitors.

Tumor Cells	Tumor Microenvironment	Circulating Factors	Host Factors	Immune-Related Adverse Events
PDL-1 expression	PDL-1 expression	Peripheral blood cells, e.g., myelogenous cells, eosinophils, macrophages, CD+ ICSO+ T cells	Age, gender, body fat distribution	Endocrine immune-related adverse events, e.g., thyroid dysfunction Skin immune-related adverse events e.g., vitiligo, pruritus, lichenoid toxicity
TMB	Tumor-infiltrating lymphocytes, e.g., CD39+ CD8+ T cells. CD4+ T cells, FOXP3+ T cells, TAMs, myeloid cells, NKp46+ cells	Other circulating factors, e.g., PDL-1, soluble proteins, cytokines and inflammatory factors	Host germline mutations, HLA diversity, and other specific mutations	
DDR pathways: dMMR/MSI	Immune status of tumor microenvironment	Circulating nucleic acids, e.g., ctDNA, RNA (mRNA, miRNA)	Intestinal commensal microbiota	
Specific gene mutations	Immunologic classification, immunoscore	Circulating tumor cells (CTCs)		
Neoantigen load	Diversity of immune cell repertoire			
	TIL richness and clonality			

Predictive biomarkers of response to ICI therapies include the expression of intermolecular interactions within tumor cells to the expression of various molecules and cells in the TME and also circulating tumor and host factors. Adapted from ref. [36]. Definitions of abbreviations are found in the abbreviations list at the end of the text.

**Table 2 ijms-24-00041-t002:** Types of biomarkers and their clinical utility in metastatic melanoma.

	Diagnostic Biomarkers	Prognostic Biomarkers	Predictive Biomarkers		On-Treatment Biomarkers
Purpose	Presence of disease	Overall patient survival independent of therapy	Response to treatment (efficacy or toxicity)		Pharmacodynamic biomarkers (drug interaction with its target)
Time of evaluation	Before diagnosis and at diagnosis	At diagnosis	Before treatment selection		During or post-treatment
Clinical utility	Before diagnosis: Allows screening of healthy patients At diagnosis: Stratifies benign vs. malignant, classify into subtypes	At diagnosis: Estimate risk of disease Post diagnosis: Allows monitoring disease status, detects recurrence	Identify treatments likely to be effective, guides initial treatment strategy and decision making		Determines degree of drug response, guides treatments decision making during treatment
Current state			Validated	Emerging	
	Human Melanoma Black-45 (HMB-45)	Lactate dehydrogenase	BRAF V600 mutation	TMB	ctDNA profiles
	Melan-A	M stage		Neoantigen load	Absolute lymphocyte count
	Tyrosinase	Disease sites		Molecular alterations	Proliferating CD8+ T cells
	Microphthalmia transcription factor (MITF)			PDL-1 expression	Increase of T-cell subsets and checkpoint molecules (PDL-1, LAG-3)
	S100			LAG-3 expression	Granzyme B expression
	SM5-1			CD8+ T cells at tumor invasive margin	T-cell receptor (TCR) signature
	Chondroitin sulfate proteoglycan 4 (CSPG4)			IL-17 expression	
	Loss of p16 protein expression			Immune-related gene expression signatures	
	Biomarker panels and gene arrays			T-cell receptor (TCR) signature	

Adapted from ref. [37,38,39,40]. Definitions of abbreviations are found in the abbreviations list at the end of the text.

**Table 3 ijms-24-00041-t003:** Genomic correlations with response and resistance based on primary location.

Primary Location	Response Category	Characteristics or Modality	References
T cell	Intratumoral infiltration	Transcriptional signatures of cytotoxic lymphocytes infiltrating the tumor core	[67,83,141,142,143,144]
	Enhanced effector function	Increased expression of PRF1, GZMA/B, CD8A, and IFNG	[82,95,145]
	Increased clonality	Ranging from 0 to 1, with 1 indicating a monoclonal population	[46,121,146]
	Greater stemness	Express chemokine receptor CXCR5 and transcription factor TCF7; lack TIM-3/CD39	[98,147]
	Reduced exhaustion	Express co-inhibitory receptor TIM-3 and ectonucleotidase CD39; lack CXCR5/TCF7	[98,147]
Tumor cell (response mechanisms)	Tumor antigens	Neoantigens, viral antigens	[7,66,69,72,131,148,149,150,151,152,153,154,155]
	Increased tumor mutation burden	Mismatch repair deficiency	[151,156,157]
	Immunogenic alterations	Inactivating mutations in SERPINB3 and SERPINB4	[158]
	Mutational signatures	Smoking, ultraviolet light, alkylating agent therapy, APOBEC	[66,159,160]
	Genomic upregulation of PD-L1	PDL1 amplification and loss of CDK4, SPOP, and CMTM4 and CMTM6	[130,161,162,163,164,165,166,167]
	Chromatin modifier loss	Inactivating mutations in PBRM1, ARID1A, and SMARCA4	[159,168,169,170]
Tumor cell (resistance mechanisms)	Tumor antigens	Cancer/testis antigens similar to self and less immunogenic	[171]
	Deficient antigen presentation	Inactivating mutations in B2M, HLA, JAK/STAT, and IFN-γ response genes	[151,159]
	Oncogenic pathways	Inactivating STK11 and PTEN mutations, WNT/β-catenin, EGFR and KRAS mutations	[67,135,172,173,174,175,176,177,178,179]
	Immune evasion alterations	Increased expression of SERPINB9	[82]
	CNAs	High levels of copy-number loss, chromosome arm and whole-chromosome CNAs	[46,73]
Tumor microenvironment	Immunosuppressive stromal cells	Transcriptional signatures of fibroblasts, endothelial cells, and TGF-β signaling	[30,67,180,181]
	Immunosuppressive immune cells	Transcriptional signatures of myeloid-derived suppressor cells and regulatory T cells	[82,142]

Adapted from ref. [113]. Definitions of abbreviations are found in the abbreviations list at the end of the text.

**Table 4 ijms-24-00041-t004:** Therapeutic strategies to turn “cold tumors” into “hot tumors”.

Therapeutic Modalities	Primary Effect Site	Main Mechanisms	References
Immune adjuvants (TLR agonists, STING agonists)	Draining lymph node	Promotes T cell priming by antigen release, antigen processing and presentation, and DC-T cell interaction	[187,188]
Oncolytic viruses			[189,190]
Chemotherapy and radiotherapy			[191,192]
Epigenetic modification inhibitors (DNMT inhibitor, HDAC inhibitor, EZH2 inhibitor)			[193,194,195]
Metabolic intervention			[196,197]
Local thermal ablation therapy (Radiofrequency ablation)			[198]
Photothermal therapy and photodynamic therapy			[26,199,200]
Magnetic hyperthermia			[201,202]
High-intensity focused ultrasound			[203,204]
Adoptive cellular therapy (TILs, CAR-T cells)	Draining lymph node	Promotes T cell expansion	[205,206,207]
Vaccines			[208]
Oncogenic pathway inhibitors	Tumor microenvironment	Promotes T cell trafficking and infiltration	[174,209,210]
Epigenetic modification inhibitors			[211,212,213]
Antiangiogenic therapy (anti-VEGF)			[102]
TGFβ inhibitors			[180,214,215]
CXCR4 inhibitors			[216,217]

Examples of therapeutic approaches to drive T cells into tumors include TLR and STING agonists [187,188], oncolytic viruses [189,190], chemotherapy and radiotherapy [191,192]], epigenetic modification inhibitors (DNMT inhibitor, HDAC inhibitor, EZH2 inhibitor) [193,194,195], metabolic intervention [196,197], local thermal ablation therapy (radiofrequency ablation) [198], photothermal therapy and photodynamic therapy [26,199,200], magnetic hyperthermia [201,202], and high-intensity focused ultrasound [203,204], which all promote T cell priming by antigen release, antigen processing and presentation, and DC-T cell interaction [124]. Meanwhile, adoptive cellular therapy (TILs, CAR-T cells) [205,206,207] and vaccines [208] promote T cell expansion [124]. On the other hand, oncogenic pathway inhibitors [174,209,210], epigenetic modification inhibitors [211,212,213], antiangiogenic therapy (anti-VEGF) [102], TGFβ inhibitors [180,214,215], and CXCR4 inhibitors promote T cell trafficking and infiltration [216,217]. Adapted from ref. [124]. Definitions of abbreviations are found in the abbreviations list at the end of the text.

**Table 5 ijms-24-00041-t005:** Representative examples of ICI therapy in combination with other drugs to overcome resistance.

Title	Interventions	Trial No	Conditions	Phase	Setting	Status as of 11-2022
Neoantigen vaccines						
A Phase I Study Combining NeoVax, a Personalized NeoAntigen Cancer Vaccine, With Ipilimumab to Treat High-risk Renal Cell Carcinoma	NeoVax + anti-CTLA-4 ipilimumab	NCT02950766	Stage III or low-volume stage IV renal cell carcinoma	I	Adjuvant/1st line	Recruiting
Neoantigen-based Personalized Vaccine Combined with Immune Checkpoint Blockade Therapy in Patients with Newly Diagnosed, Unmethylated Glioblastoma	NeoVax + anti-PD-1 nivolumab ± anti-CTLA-4 ipilimumab	NCT03422094	Newly diagnosed, unmethylated glioblastoma	I	1st line	Terminated (Manufacturer changed focus to cell therapy)
Personalized Neoantigen Cancer Vaccine + Pembrolizumab After Rituximab for Follicular Lymphoma	NeoVax + anti-CD20 rituxumab	NCT03361852	Follicular lymphoma	I	1st line	Recruiting
A Personalized Neoantigen Cancer Vaccine in Treatment Naïve, Asymptomatic Patients with IGHV Unmutated CLL	NeoVax + alkylating agent cyclophosphamide	NCT03219450	Chronic lymphocytic leukemia	I	1st line	Recruiting
Pooled Mutant KRAS-Targeted Long Peptide Vaccine Combined with Nivolumab and Ipilimumab for Patients with Resected MMR-p Colorectal and Pancreatic Cancer	KRAS peptide vaccine, nivolumab, ipilimumab	NCT04117087	Colorectal cancer, pancreatic cancer	I	Recurrent or metastatic	Recruiting
NeoVax Plus Ipilimumab in Renal Cell Carcinoma	NeoVax, ipilimumab	NCT02950766	Kidney cancer	I	1^st^ line	Recruiting
Immunotherapy Combined with Radiation and Influenza Vaccine for Pancreatic Cancer	Nivolumab, ipilimumab, influenza vaccine, stereotactic body radiation therapy	NCT05116917	Pancreatic cancer	II	Recurrent	Recruiting
Combination Therapy with Nivolumab and PD-L1/IDO Peptide Vaccine to Patients with Metastatic Melanoma	Nivolumab, PD-L1/IDO peptide vaccine	NCT03047928	Metastatic melanoma	I/II	Recurrent	Recruiting
A Phase I/II Trial to Evaluate a Peptide Vaccine Plus Ipilimumab in Patients with Melanoma	Ipilimumab, peptide vaccine 6MHP	NCT02385669	Melanoma	I/II	Stage III or IV, recurrent or metastatic, stage IIA, IIB-IV resected to no evidence of disease	Terminated, has Results
Vaccine Combining Multiple Class I Peptides and Montanide ISA 51VG With Escalating Doses of Anti-PD-1 Antibody Nivolumab or Ipilimumab with Nivolumab For Patients with Resected Stages IIIC/IV Melanoma	NY-ESO-1 157-165 (165 V), nivolumab, gp100:280-288 (288 V), montanide ISA 51 vegetable grade (VG), ipilimumab	NCT01176474	Melanoma	I	Stages IIIC/IV melanoma, with no evidence of disease	Active, not recruiting
Dendritic cell vaccines						
Combination Immunotherapy-Ipilimumab-Nivolumab-Dendritic Cell p53 Vac - Patients with Small Cell Lung Cancer	Dendritic cell-based p53 vaccine + nivolumab + ipilimumab	NCT03406715	Small cell lung cancer	II	Recurrent	Active, not recruiting
Nivolumab with DC Vaccines for Recurrent Brain Tumors	Dentritic cell + nivolumab	NCT02529072	Malignant glioma, astrocytoma, glioblastoma	I	Recurrent	Completed, has results
Neoantigen Dendritic Cell Vaccine and Nivolumab in HCC and Liver Metastases from CRC	Neoantigen dendritic cell vaccine + nivolumab	NCT03782064	Hepatocellular carcinoma, hepatocellular cancer, colorectal cancer, colorectal carcinoma, liver metastases	II	Newly diagnosed or recurrent hepatocellular carcinoma	Recruiting
Dendritic Cell (DC)/Myeloma Fusions in Combination with Nivolumab in Patients with Relapsed Multiple Myeloma		NCT03782064	Multiple myeloma	II	Relapsed	Terminated Has Results
Polarized Dendritic Cell (aDC1) Vaccine, Interferon Alpha-2, Rintatolimod, and Celecoxib for the Treatment of HLA-A2 + Refractory Melanoma	Alpha-type-1 polarized dendritic cells, celecoxib, PD-L1 inhibitor, PD-1 inhibitor, recombinant Interferon alpha-2b, rintatolimod	NCT04093323	HLA-A2 positive cells present, refractory melanoma	II	Refractory	Recruiting
Interferon therapies						
Testing the Combination of Two Experimental Drugs MK-3475 (Pembrolizumab) and Interferon-gamma for the Treatment of Mycosis Fungoides and Sézary Syndrome and Advanced Synovial Sarcoma	Interferon-γ-1β + anti-PD-1 pembrolizumab	NCT03063632	Mycosis fungoides and Sézary syndrome	II	Refractory	Active, not recruiting
Combination of Interferon-gamma and Nivolumab for Advanced Solid Tumors	Interferon-γ + anti-PD-1 nivolumab	NCT02614456	Advanced solid tumors	I	2nd line	Completed
Pembrolizumab Combined with Itacitinib (INCB039110) and/or Pembrolizumab Combined with INCB050465 in Advanced Solid Tumors	JAK1 inhibitor itacitinib or PI3Kδ inhibitor + anti-PD-1 pembrolizumab	NCT02646748	Advanced solid tumors	I/II	Refractory	Completed
Pembrolizumab and Ruxolitinib Phosphate in Treating Patients with Metastatic Stage IV Triple Negative Breast Cancer	JAK2 inhibitor ruxolitinib + anti-PD-1 pembrolizumab	NCT03012230	Metastatic triple-negative breast cancer	I	2nd line	Recruiting
Pembrolizumab and Itacitinib (INCB039110) for Non-Small Cell Lung Cancer	JAK1 inhibitor itacitinib + anti-PD-1 pembrolizumab	NCT03425006	Metastatic PD-L1^+^ non-small cell lung cancer	II	1st line	Terminated
Study of the Safety and Efficacy of MIW815 with PDR001 in Patients with Advanced/Metastatic Solid Tumors or Lymphomas	STING agonist MIW815 + anti-PD-1 spartalizumab	NCT03172936	Advanced solid tumors or lymphomas	I	Any line	Terminated (Sponsor’s decision)
Trial of Intratumoral Injections of TTI-621 in Subjects with Relapsed and Refractory Solid Tumors and Mycosis Fungoides	TTI-621 monotherapy, TTI-621 + PD-1/PD-L1 inhibitor, TTI-621 + pegylated interferon-α2aOther: TTI-621 + T-Vec, TTI-621 + radiation TTI-621 (SIRPα-IgG1 Fc) anti- CD47 “don’t eat me” signal	NCT02890368	Solid tumors, mycosis fungoides, melanoma, merkel-cell carcinoma, squamous cell carcinoma, breast carcinoma, human papillomavirus-related malignant neoplasm, soft tissue sarcoma	I	Relapsed and refractory	Terminated
A Study to Assess the Safety and Tolerability of Atezolizumab in Combination with Other Immune-Modulating Therapies in Participants with Locally Advanced or Metastatic Solid Tumors	Atezolizumab, bevacizumab, interferon alpha-2b, ipilimumab, obinutuzumab, PEG-interferon alpha-2a	NCT02174172	Solid tumors	I	Locally advanced or metastatic solid tumors	Completed
The Study of JS001 Compared to High-Dose Interferon in Patients with Mucosal Melanoma That Has Been Removed by Surgery	Humanized anti-PD-1 monoclonal antibody toripalimab, high-dose recombinant interferon a-2B	NCT03178123	Mucosal melanoma	II	1st line after surgery	Active, not recruiting
Ipilimumab or High-Dose Interferon Alfa-2b in Treating Patients with High-Risk Stage III–IV Melanoma That Has Been Removed by Surgery	Ipilimumab, interferon alpha-2b	NCT01274338	Melanoma of unknown primary, recurrent melanoma, stage IIIB cutaneous melanoma AJCC v7, stage IIIC cutaneous melanoma AJCC v7, stage IV cutaneous melanoma AJCC v6 and v7	III	Recurrence after surgery	Active, not recruiting, has results
Ipilimumab With or Without High-Dose Recombinant Interferon Alfa-2b in Treating Patients with Stage III–IV Melanoma That Cannot Be Removed by Surgery	Ipilimumab, recombinant interferon alpha-2b	NCT01708941	Recurrent melanoma, stage IIIA cutaneous melanoma AJCC v7, stage IIIB cutaneous melanoma AJCC v7, stage IIIC cutaneous melanoma AJCC v7, stage IV cutaneous melanoma AJCC v6 and v7	II	Unresectable stage III or stage IV melanoma, either initial presentation or recurrent	Active, not recruiting, has results
Nivolumab, Fluorouracil, and Interferon Alpha 2B for the Treatment of Unresectable Fibrolamellar Cancer	Fluorouracil, nivolumab, recombinant interferon alpha 2b-like protein	NCT04380545	Stage III IVB epatocellular carcinoma AJCC v8, unresectable fibrolamellar carcinoma	I/II	1st line	Recruiting
High-Dose Recombinant Interferon Alfa-2B, Ipilimumab, or Pembrolizumab in Treating Patients with Stage III–IV High Risk Melanoma That Has Been Removed by Surgery	Ipilimumab, pembrolizumab, recombinant interferon alpha-2b	NCT02506153	Cutaneous melanoma, metastatic mucosal melanoma, metastatic non-cutaneous melanoma, non-cutaneous melanoma, recurrent cutaneous melanoma, recurrent mucosal melanoma, recurrent non-cutaneous melanoma, and stage III-IVC of all above	III	1st line, high risk melanoma that has been removed by surgery	Active, not recruiting
Safety and Tolerability of Pembrolizumab (MK-3475) + Pegylated Interferon Alfa-2b and Pembrolizumab+ Ipilimumab in Participants with Advanced Melanoma or Renal Cell Carcinoma (MK-3475-029/KEYNOTE-29)	Pembrolizumab, PegIFN-2b, Ipilimumab	NCT02089685	Renal cell carcinoma, melanoma	I/II	Advanced or unresectable or metastatic renal cell carcinoma, melanoma	Completed Has Results
IPI-Biotherapy for Patients Previously Treated with Cytotoxic Drugs with Metastatic Melanoma	Ipilimumab, interferon, interleukin-2 (aldesleukin)	NCT01409187	Melanoma	I/II	Recurrent	Withdrawn
TIL and Anti-PD1 in Metastatic Melanoma	Nivolumab & tumor- infiltrating lymphocytes with/without interferon-alpha	NCT03638375	Melanoma	I/II	Recurrent	Recruiting
Trial of Intratumoral Injections of TTI-621 in Subjects with Relapsed and Refractory Solid Tumors and Mycosis Fungoides	TTI-621 (SIRPα-IgG1 Fc) monotherapy TTI-621 + PD-1/PD-L1 inhibitor, TTI-621 + pegylated interferon-α2a, TTI-621 + T-Vec, TTI-621 + radiation	NCT02890368	Solid tumors, mycosis fungoides Melanoma, merkel-cell carcinoma, squamous cell carcinoma, breast carcinoma, human papillomavirus-related malignant neoplasm, soft tissue sarcoma	I	Recurrent	Terminated
CDK4/6 Inhibitors						
Palbociclib After CDK and Endocrine Therapy (PACE)	Fulvestrant ± CDK4/6 inhibitor palbociclib ± anti-PD-L1 avelumab	NCT03147287	Metastatic HR^+^HER2^−^ breast cancer	II	2nd–3rd line	Active, not recruiting
Neoadjuvant Endocrine Therapy, Palbociclib, Avelumab in Estrogen Receptor Positive Breast Cancer (ImmunoADAPT)	Tamoxifen ± CDK4/6 inhibitor palbociclib followed by anti-PD-L1 avelumab	NCT03573648	Stage II or III HR^+^ breast cancer	II	Neoadjuvant	Recruiting
Ribociclib + PDR001 in Breast Cancer and Ovarian Cancer	CDK4/6 inhibitor ribociclib + anti-PD-1 spartalizumab ( + fulvestrant if breast)	NCT03294694	Metastatic ovarian cancer or HR^+^HER2^−^ breast cancer	I	Any line	Terminated (safety implications)
A Study of Abemaciclib (LY2835219) in Participants with Non-Small Cell Lung Cancer or Breast Cancer	CDK4/6 inhibitor abemaciclib + anti-PD-1 pembrolizumab	NCT02779751	Metastatic non-small cell lung cancer or HR^+^HER2^−^ breast cancer	I	1st–3rd line	Active, not recruiting
Avelumab, Cetuximab, and Palbociclib in Recurrent or Metastatic Head and Neck Squamous Cell Carcinoma	Anti-PD-L1 avelumab, cetuximab, + CDK4/6 inhibitor palbociclib	NCT03498378	Recurrent/metastatic head and neck squamous cell carcinoma	I	Any line	Active, not recruiting
Carboplatin, Etoposide, and Atezolizumab with or Without Trilaciclib (G1T28), a CDK4/6 Inhibitor, in Extensive-Stage SCLC	Anti-PDL-1 atezolizumab, trilaciclib, carboplatin, etoposide, placebo	NCT03041311	Small cell lung cancer	II	1st line	Terminated, has results
Avelumab or Hydroxychloroquine with or Without Palbociclib to Eliminate Dormant Breast Cancer	Anti-PDL-1 avelumab, HCQ, palbociclib	NCT04841148	Breast cancer	II	Refractory	Recruiting
Phase Ib Study of TNO155 in Combination with Spartalizumab or Ribociclib in Selected Malignancies	Anti-PD-1 spartalizumab, ribociclib, TNO155	NCT04000529	Non-small cell lung cancer, head and neck squamous cell carcinoma, esophageal squamous cell carcinoma, gastrointestinal stromal tumors, colorectal cancer	I	Refractory	Recruiting
Ribociclib and Spartalizumab in R/M HNSCC	Anti-PD-1 spartalizumab, ribociclib,	NCT04213404	Head and neck squamous cell carcinoma	I	Recurrent	Active, not recruiting
Anti-TGF-β therapies						
Phase I/Ib Study of NIS793 in Combination with PDR001 in Patients with Advanced Malignancies	Anti-TGF-β NIS793 + anti-PD-1 spartalizumab	NCT02947165	Advanced solid tumors	I	Refractory	Completed
A Study of Galunisertib (LY2157299) and Durvalumab (MEDI4736) in Participants with Metastatic Pancreatic Cancer	TGF-β receptor 1 inhibitor galunisertib + anti-PD-L1 durvalumab	NCT02734160	Metastatic pancreatic cancer	I	3rd line	Completed
A Study of Galunisertib (LY2157299) in Combination with Nivolumab in Advanced Refractory Solid Tumors and in Recurrent or Refractory NSCLC, or Hepatocellular Carcinoma	TGF-β receptor 1 inhibitor galunisertib + anti-PD-1 nivolumab	NCT02423343	Recurrent or refractory non-small cell lung cancer or hepatocellular carcinoma	I/II	2nd line	Completed
MSB0011359C (M7824) in Metastatic or Locally Advanced Solid Tumors	Anti-PD-L1/TGF-β trap M7824	NCT02517398	Metastatic or locally advanced solid tumors	I	Refractory	Completed
M7824 in Treating Patients with Stage II–III HER2 Positive Breast Cancer	Anti-PD-L1/TGF-β trap M7824 + treatment of physician’s choice	NCT03620201	Stage II–III HER2^+^ breast cancer	I	Neoadjuvant	Active, not recruiting
M7824 vs. Pembrolizumab as a First line (1L) Treatment in Participants with Programmed Death-ligand 1 (PD-L1) Expressing Advanced Non-small Cell Lung Cancer (NSCLC)	Anti-PD-L1/TGF-β trap M7824 vs. anti-PD-1 pembrolizumab	NCT03631706	PD-L1^+^ non-small cell lung cancer	II	1st line	Active, not recruiting
Antitumor Activity of Vactosertib in Combination with Pembrolizumab in Acral and Mucosal Melanoma Patients Progressed from Prior Immune Check Point Inhibitor	Anti-PD-1 pembrolizumab + anti-TGF-βR1 vactosertib	NCT05436990	Acral melanoma, mucosal melanoma	II	Relapsed	Not yet recruiting

Examples of recent clinical trials investigating various novel therapeutic modalities as interventions in combination with ICI to overcome monotherapy resistance and attempt to turn “cold tumors’ into “hot tumors”. Adapted from ref. [113]. Definitions of abbreviations are found in the abbreviations list at the end of the text.

## Data Availability

The authors grant the publisher the sole and exclusive license of the full copyright in the Contribution, which license the publisher hereby accepts.

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
