# Peer review of "Insights and Strategies of Melanoma Immunotherapy: Predictive Biomarkers of Response and Resistance and Strategies to Improve Response Rates"

_ijms, 2022, doi:10.3390/ijms24010041_

Round 1

Reviewer 1 Report

this is an impressive review on predictive biomarkers of response to immunotherapy in melanoma. I feel that it is a major addition to the current bulk of knowledge in the field. The review flows very well and charts are really useful.

I have only one comment. I understand that this is a narrative review. However, can the Authors comment on how literature was browsed to identify the evidence commented upon in the paper?

Author Response

Response to reviewer 1: We thank for the reviewer for the very helpful comment. Below is our point-by-point response to each comment (in red).

Response 1. We have reviewed recent research and review articles primarily using PubMed as well as other search engines including Google to identify recent publications focusing on immune checkpoint inhibitors (ICI), melanoma biomarkers of response to ICI therapy with precise questions in mind including, the current therapies but specifically ICI therapies for melanoma, molecular mechanisms underlining the response and resistance to ICI therapies, including the complex tumor microenvironment (TME), tumor-immunity and resistance mechanisms involving tumor immunophenotypes, predictive biomarkers including current and emerging genomic and gene expression correlates of ICI response including those related to neoantigens, antigen presentation, DNA repair, and oncogenic pathways, immune checkpoint regulators, T cell functionality, chromatin modifiers, and copy-number alterations in mediating selective response to ICI, and to stratify patients to targeted treatments  including ICI alone or in combination with other emerging therapies, and novel therapeutic strategies to turn “cold tumors” into “hot tumors” including the combination therapy strategies with ICI, and examples of those combinations currently in clinical trials.

We also paid close attention to the data quality and integrity of the data and if the data were technically sound and if the papers reviewed provide strong evidence for their conclusions, and if some of those findings or approaches were translatable i.e., whether they have been tested in clinical trials or approved by the FDA.

Reviewer 2 Report

The manuscript is very interesting. The Authors have reviewed a great range of studies melanoma therapy with immune checkpoint inhibitors. A lot of interesting information was presented in the manuscript. The information is described clearly in the text, summarized in tables, or presented as figures. I would like to emphasize that the figures have good quality and definitely increase the value of the manuscript.

I recommend the manuscript for publication after making appropriate revisions according to the following comments.

1.      There are many punctuation errors in the manuscript. Here is an example of one such error: „…for patients with advanced melanoma. has been rather unreliable [22].” (p.2)

2.      Many references can be found in the inappropriate place in the manuscript.

a.      „Melanoma affects more than 1 million Americans, and there is an increasing incidence of melanoma worldwide, approx. 300,000 new cases are diagnosed in the US each year [5] [5],..”(p.1)

b.      Numerous successes have been achieved with anti-PD1, anti-CTLA4, or combination therapies [5] [5,7] …” (p.2)

c.      „Du et al [60] reported that several genomic and transcriptomic-based biomarkers have been explored as potential predictors of ICI response [7,31,42,61-73].” (p.7) - the Authors refer to the Duke et al. publication and introduce many more references.

d.      „Unfortunately, many of those potential biomarkers of ICI response have not been validated yet. [67,82-85].”(p.7) -  in my opinion, the Authors should introduce more information from the indicated references

e.      „Tian et al [90] reported that combined BRAF, KRAS, and PI3KCA mutation signatures resulted in a favorable predictive response to cetuximab for patients with colorectal cancer [90].” (p.8)

f.       „Contrary to that, Chen et al. have developed adaptive immune signatures based on tumor samples obtained during the early course of treatment showing that those signatures were highly predictive of response to ICI in patients with metastatic melanoma [102].” - The 102 reference is not Chen et al.publication

g.      „Other sources of immunogenic antigens including immunogenic epitopes can also derive from mutations associated with gene fusion, aberrant messenger-RNA splicing with retained introns, or aberrant translation resulting in cryptic antigens [106]. Furthermore, genomically integrated endogenous retroviral sequences as a result of previous retroviral infections, although they are epigenetically silenced, can be reactivated in tumors [106], as in the case of cancer germline antigens” (p.9) - duplication of the same reference.

h.      „Grasso et al. [92] have demonstrated that immunotherapies against immune checkpoints (e.g., CTLA-4 or the PD-1) down regulate two main negative regulators of the antitumor immune responses [115-117] resulting in durable anti-tumor responses in a subset of cancer patients including melanoma [2,118].”(p.10) - the Authors refer to the Grasso et al. publication and introduce many more references.

i.       „Other key factors contributing to anti-tumor immune response following ICI treatment include [92] the pre-existing level of T cell infiltration of the tumor [119-121] representing the immunogenicity of the cancer cells.” (p.10) - in my opinion, too many references.

j.       „Liu et al demonstrated that the response to anti-PD-1 therapy (with or without prior anti-CTLA-4 treatment) was associated with increased MHC-I and MHC-II expression [72]. This study demonstrated that patients not responding to therapy have occasional genetic alterations in antigen presentation genes[72]” (p.10)

k.      In the description of Figures 3 and 4 there are references to publications based on which the figures were not prepared.

l.       There are 2 contradictory pieces of information in the description of Figure 5: „Adapted from [113].” and ” Adapted from [113,159,160].”

m.    In my opinion, the description of Table 4 repeats the information in the table and therefore is not needed.

n.      „In addition, several other key challenges facing ICI therapy must be addressed to move the field ahead [155] including (1) development of preclinical models that translate to human immunity (2), identification and validation of the dominant drivers involving cancer immunity, (3) deeper insights into organ-specific immune TME, (4) insights into the differences of molecular and cellular drivers between primary and secondary immune escape, (5) characterization of the benefit of endogenous versus synthetic immunity, (6) investigation of ICI therapy combinations with other drugs (Figure 6, Table 4 and 5) in early-phase clinical studies, (7) investigation of steroids and immune suppression on immunotherapy and autoimmune adverse events and toxicities, (8) boosting the personalized medicine approaches by using composite biomarkers, (9) developing and implementing refined regulatory endpoints for immunotherapy, and (10) optimizing durable survival with a combination of multi-agent immunotherapy regimens [155].”(p.29)

3.      The titles of Tables 1 and 2 are repeated in the first rows of the tables.

4.      The " Current state - Validated - Emerging " row of Table 2 makes the down side of the table poorly understood.

5.      The following fragments are most probably the titles of subsequent parts of the manuscript, but they are not properly marked:

a.      Gene expression signatures at baseline and on-treatment tumor specimens (p.8)

b.      Tumor antigens (p.9)

c.      Genomic alterations (p.10)

Author Response

POINT BY POINT RESPONSE TO REVIEWER'S COMMENTS:

We thank the reviewers for their comments, which were very helpful in revising the manuscript to convey our message more clearly our work. We provide below detailed responses to each comment and the appropriate changes.

Reviewer 2.

Comments and Suggestions for Authors

The manuscript is very interesting. The Authors have reviewed a great range of studies melanoma therapy with immune checkpoint inhibitors. A lot of interesting information was presented in the manuscript. The information is described clearly in the text, summarized in tables, or presented as figures. I would like to emphasize that the figures have good quality and definitely increase the value of the manuscript.

I recommend the manuscript for publication after making appropriate revisions according to the following comments.

Authors' response to the reviewer 2 Comments

Response to reviewer 2: We thank the reviewer for the helpful comments. Below is our point-by-point response to each comment (in red).

  1. There are many punctuation errors in the manuscript. Here is an example of one such error: „…for patients with advanced has been rather unreliable [22].” (p.2)

RESPONSE: Corrected...

  1. Many references can be found in the inappropriate place in the manuscript.

  1. „Melanoma affects more than 1 million Americans and there is an increasing incidence of melanoma worldwide, approx. 300,000 new cases are diagnosed in the US each year [5] [5],..”(p.1)

RESPONSE: Corrected...

  1. Numerous successes have been achieved with anti-PD1, anti-CTLA4, or combination therapies [5] [5,7]…” (p.2)

RESPONSE: Corrected...

  1. „Du et al [60] reported that several genomic and transcriptomic-based biomarkers have been explored as potential predictors of ICI response [7,31,42,61-73].” (p.7) - the Authors refer to the Duke et al.publication and introduce many more references.

RESPONSE: Corrected...

  1. „Unfortunately, many of those potential biomarkers of ICI response have not been validated yet. [67,82-85].”(p.7) - in my opinion, the Authors should introduce more information from the indicated references

RESPONSE: Corrected.. as follows:

A recent study by Carter et al. [75] questioned the validity of immuno-predictive score (IMPRES), a predictor of ICI response in melanoma consisting of  15 pairwise transcriptomic signatures analyzing the relations between immune checkpoint gene reported by Auslander et al. [77]. The IMPRES is context-dependent and could not reproducibly predict ICI response in the context of metastatic melanoma  [77].

Moreover, Xiao et al. [78] have questioned the reproducibility of immune cells.Sig [80] across different RNAseq datasets demonstrating inconsistency of prediction capability of ImmuneCells.Sig across different RNA-seq datasets [78].  The performance of the ImmuneCells.Sig signature on predicting ICI outcomes in four melanoma patient datasets set by using the same implementation scheme with Xiong et al. [80] showed that there was inconsistency across different datasets [78].

  1. „Tian et al [90] reported that combined BRAF, KRAS,and PI3KCA mutation signatures resulted in a favorable predictive response to cetuximab for patients with colorectal cancer [90].” (p.8)

  1. „Contrary to that, Chen et al. have developed adaptive immune signatures based on tumor samples obtained during the early course of treatment showing that those signatures were highly predictive ofresponse to ICI in patients with metastatic melanoma[102].” - The 102 reference is not Chen et al publication _

RESPONSE: Changed to Wallin et al

  1. J. Wallin, J. C. Bendell, R. Funke, M. Sznol, K. Korski, S. Jones, et al. Atezolizumab in combination with bevacizumab enhances antigen-specific T-cell migration in metastatic renal cell carcinoma. Nat Commun 2016 Vol. 7 Pages 12624. Accession Number: 27571927 PMCID: PMC5013615 DOI: 10.1038/ncomms12624

  1. „Other sources of immunogenic antigens including immunogenic epitopes can also derive from mutations associated with gene fusion, aberrant messenger-RNA splicing with retained introns, or aberrant translation resulting in cryptic antigens [106]. Furthermore, genomically integrated endogenous retroviral sequences as a result of previous retroviral infections, although they are epigenetically silenced can be reactivated in tumors [106], as in the case of cancer germline antigens” (p.9) - duplication of the same reference.

RESPONSE: Corrected.. (eliminated 1st citation of [106] and consolidated in only one reference in one sentence as shown in the text and below.

Other sources of immunogenic antigens including immunogenic epitopes can also derive from mutations associated with gene fusion, aberrant messenger-RNA splicing with retained introns, or aberrant translation resulting in cryptic antigens, genomically integrated endogenous retroviral sequences as a result of previous retroviral  infections, although they are epigenetically silenced can be reactivated in tumors [106], as in the case of cancer germline antigens” (p.9

  1. „Grasso et al. [92] have demonstrated that immunotherapies against immune checkpoints (e.g.,CTLA-4 or the PD-1) down regulate two main negative regulators of the antitumor immune responses [115-117] resulting in durable anti-tumor responses in a subset of cancer patients including melanoma [2,118].”(p.10) - the Authors refer to the Grasso et al. publication and introduce many more references.

RESPONSE: Corrected.. in the manuscript as following

Recent data  have demonstrated that immunotherapies against immune checkpoints (e.g., CTLA-

4 or the PD-1) down regulate two main negative regulators of the anti tumor immune responses [

94][116-118] resulting in durable anti-tumor responses in a subset of cancer patients including

melanoma [2, 119].

  1. „Other key factors contributing to anti-tumor immuneresponse following ICI treatment include [92] the pre-existing level of T cell infiltration of the tumor [119-121] representing the immunogenicity of the cancer cells.” (p.10) - in my opinion, too many references.

RESPONSE: We agree with the reviewer’s point; however, several references support the reported findings.

  1. „Liu et al demonstrated that the response to anti-PD-1therapy (with or without prior anti-CTLA-4 treatment)was associated with increased MHC-I and MHC-II expression [72]. This study demonstrated that patients not responding to therapy have occasional genetic alterations in antigen presentation genes[72]”(p.10)

  1. In the description of Figures 3 and 4 there are references to publications based on which the figures were not prepared.

RESPONSE: we have removed References in the Figure legends but retained those that were the source of the Figures that we have adapted from.

  1. There are 2 contradictory pieces of information in the description of Figure 5: „Adapted from [113].” and ”Adapted from [113,159,160].”

  1. In my opinion, the description of Table 4 repeats the information in the table and therefore is not needed.

RESPONSE: The information under Table 4 is in fact a paragraph (not a legend), underlying the relevance of the information in Table 4.

  1. „In addition, several other key challenges facing ICI therapy must be addressed to move the field ahead[155] including (1) development of preclinical models that translate to human immunity (2), identification and validation of the dominant drivers involving cancer immunity, (3) deeper insights into organ-specific immune TME, (4) insights into the differences of molecular and cellular drivers between primary and secondary immune escape, (5) characterization of the benefit of endogenous versus synthetic immunity, (6) investigation of ICI therapy combinations with other drugs (Figure 6, Table 4 and 5) in early-phase clinical studies, (7) investigation of steroids and immune suppression on immunotherapy and autoimmune adverse events and toxicities, (8) boosting the personalized medicine approaches by using composite biomarkers, (9) developing and implementing refined regulatory endpoints for immunotherapy, and (10) optimizing durable survival with a combination of multi-agent immunotherapy regimens [155].”(p.29)

RESPONSE: The above text is in the Conclusion and Future Directions section and does not repeat the information that is I Table 4.

  1. The titles of Tables 1 and 2 are repeated in the first rows of the tables.

RESPONSE: Corrected..

  1. The " Current state - Validated - Emerging " row of Table 2 makes the downside of the table poorly understood.

RESPONSE: In the original submission, the formatting of Table 2 was as shown below, which we believe is clear. However, the formatted version sent out might be misaligned.

  1. The following fragments are most probably the titles of subsequent parts of the manuscript, but they are not properly marked:

  1. Gene expression signatures at baseline and on-treatment tumor specimens (p.8)

  1. Tumor antigens (p.9)

  1. Genomic alterations (p.10)

RESPONSE: We have reformatted the subheadings accordingly. We apologize for the confusion that this might have generated.

Submission Date

19 November 2022

Date of this review

07 Dec 2022 19:51:28
